# Seismic Interaction between Multistory Pilotis RC Frames and Shorter Structures with Different Story Levels—Floor-to-Column Pounding

Grigorios E. Manoukas and Chris G. Karayannis *

Civil Engineering Department., Aristotle University of Thessaloniki, 54124 Thessaloniki, Greece; grman@civil.auth.gr
* Correspondence: karayannis@civil.auth.gr

**Abstract:** Structural pounding between adjoining multistory buildings with different total heights and different story levels has been repeatedly identified as a frequent cause of severe damage during seismic excitations. This phenomenon is very intense when upper floor slabs of short buildings hit the columns of taller and more flexible structures within their deformable length. On the other hand, it is well accepted that infill masonry panels strongly affect the seismic response and overall behavior of multistory reinforced concrete (RC) frames and especially in the common case of an open first story (pilotis). Thereupon, the interaction between a multistory frame with an open first floor and shorter and stiffer adjacent buildings was studied and the influence of the open first story on pounding investigated with inelastic dynamic step-by-step analyses. The results of the pounding cases of an 8-story RC frame with a single story and 4-story buildings were examined. Three cases of short structures were considered as follows: a frame structure, a stiff structure and a very stiff non-self-vibrating one. All studied interaction cases included type A (floor-to-floor) pounding cases and type B (floor-to-column) pounding cases. This study focused on the influence of an open first story (pilotis) on the pounding phenomenon. Therefore, all examined two-building poundings were studied considering two cases: the first case involving a fully infilled 8-story frame and second case involving an infilled 8-story structure with an open first story (pilotis). Moreover, as expected due to the asymmetry of the examined two-structured pounding pairs, the directions (plus and minus) of the seismic excitation proved to be important for the evaluation of the developing capacity demands. In the present study for the first time, it is stressed that pounding cases between structures with different geometries (asymmetric) have to be examined in both directions (plus and minus) of each seismic excitation. From the results, it can be deduced that the developing shear forces on the columns that suffer a hit in the case of type B pounding exceed the shear strength of the column even if detailing for critical regions according to Eurocode 8 is applied. Further, it is inferred that pilotis configuration increases the developing pounding forces and consequently increases the capacity demands mainly in terms of the ductility of the column that suffers the hit.

**Keywords:** structural pounding; floor-to-column pounding; influence of open first story; pilotis; reinforced concrete multistory frames

## 1. Introduction

Based on field observations after destructive earthquakes and on essential conclusions that have been drawn so far through numerous reports, it has become widely accepted that interaction between adjacent buildings is a common cause of damage during seismic excitations [1]. This phenomenon between buildings or even parts of the same building in contact or in close proximity to each other is also known as structural pounding. Pounding has been routinely reported by earthquake investigators over the past several decades. From the substantial knowledge that has been acquired, it is evident that structural pounding is

always observed when earthquakes strike densely populated areas or centers of big cities. Observations regarding severe damages due to pounding were first reported after the 1964 Alaska earthquake and the 1972 Managua earthquake [1]. Further, in the case of the earthquake in Mexico in 1985, the first assessment after in situ observations revealed that a big part (almost 40%) of the observed damage was attributable to structural pounding, as reported by Rosenblueth and Meli (1985) [2] and Bertero (1986) [3]. According to this assessment that was subsequently partially revised, a significant part of collapse cases was also attributable to the interaction between structures. Furthermore, the earthquake of Loma Prieta (1989) indicated the significant seismic hazard of structural pounding since many cases of pounding were reported over a wide distance of almost 90 km far from the epicenter in the populated urban Bay Area in both San Francisco and Oakland [4,5]. Although the earthquakes of Mexico City and Loma Prieta are unique seismic events in terms of damage and collapse cases attributed to the interaction of structures and it seems that the phenomenon has been overestimated and overstated concerning consequent structural damage and collapse, it is a fact that structural pounding is always present in all major earthquakes. Thereupon, many cases of moderate or major structural interaction damage were also identified in the seismic events of Aegion Greece 1995, Kalamata Greece 1986 (Figure 1a) and Alkyonides Greece 1981 (Figure 1b).

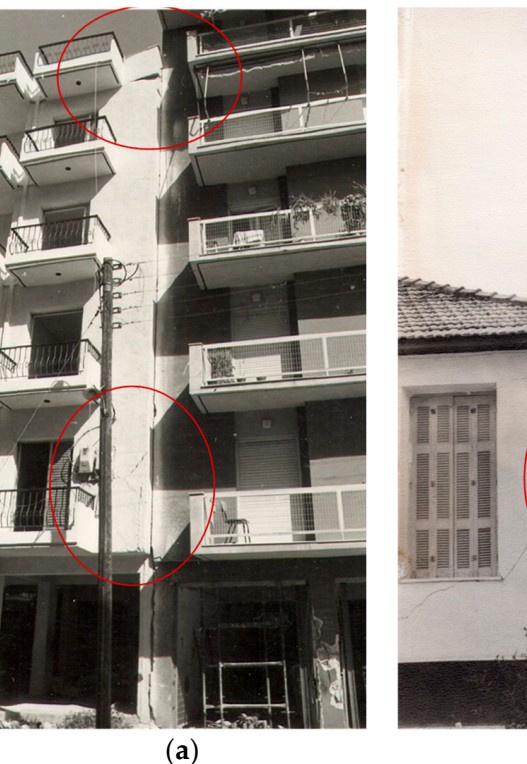 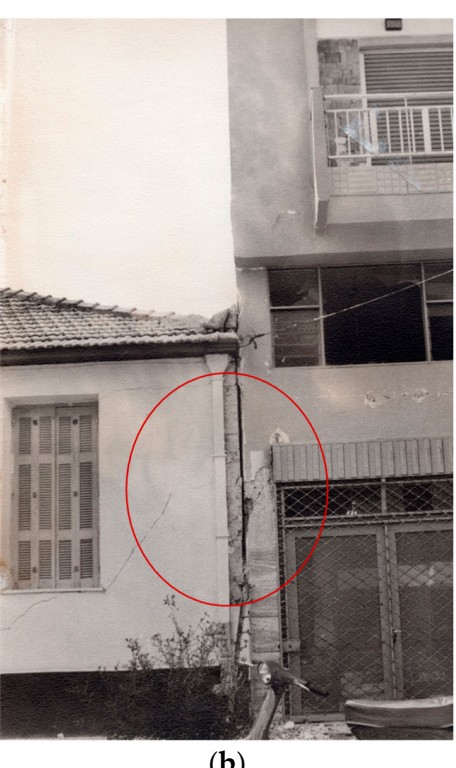

(**a**)  (**b**)

**Figure 1.** Interaction between adjacent structures. Many cases of structural interaction showing moderate damage and major damage identified after various seismic events in Greece. Type A and type B pounding. (**a**) Type A pounding between two multistory RC pilotis buildings (Kalamata 1986). (**b**) Type B pounding between a multistory fully infilled RC building and a stiff single-story house (Alkyonides 1981).

Contemporary seismic codes such as Eurocode 8 [6] specify certain separation gaps between adjacent buildings in order to avoid structural interaction. Nevertheless, a huge number of existing structures have been built in contact or in close proximity to their adjoining structures. Moreover, it is noted that separation gaps as specified by codes are not directly connected to the seismic hazard or the local capacity of the structural elements that may suffer the pounding effect. These neglected issues may occasionally lead to separations that are not adequate or consistent with the philosophy of modern seismic regulations that

allow large deformations and consequently large lateral displacements in the design during seismic excitation since they are based on the vital assumption of inelastic response.

In view of these observations, aspects of the problem of earthquake-induced structural interaction between adjacent buildings have been investigated for almost three decades. The first studies were focused on the pounding of interacting, single degree of freedom systems during earthquake excitations [7]; afterwards, the interaction of multistory reinforced concrete structures was studied towards an accurate evaluation of the structural pounding hazard. In these studies, the colliding multistory structures had equal story heights and therefore collisions took place between the floors of the interacting buildings [8–10]. Structural pounding may lead to increased shear forces, increased floor accelerations and increased interstory drifts. Further, it produces high-impact forces between the interacting structures while the floor displacements and the upper floor's total displacement may significantly increase. The amplification level of the response parameters of the interacting buildings depends on the dynamic characteristics of the colliding structures [8,9] and whether these structures are in contact or there is a separation gap between them; in this case, a very important parameter is the size of the existing gap [11].

The identification and studies of the parameters of the pounding phenomenon have been reported so far. Karayannis and Fotopoulou (1998) [11] investigated the pounding problem between reinforced concrete multistory frames designed according to contemporary seismic codes. They examined various cases of structural interaction between multistory structures designed according to Eurocodes 2 and 8 and presented initial results about the influence of pounding on the ductility demands of the columns. In these studies, the story levels of the interacting structures were always at the same heights and therefore the hits always took place between slabs (type A pounding). Karayannis and Favvata [12,13] for the first time examined the pounding problem between reinforced concrete multistory frames with different numbers of stories and further, the story levels of the first structure were different from the story levels of the other one. Consequently, the slabs of the floors of each structure hit columns of the other; this kind of pounding is referred to as type B pounding in these studies or floor-to-column pounding in other recent works. Extensive work on this type of pounding has revealed that the shear demand and local ductility demand of the columns of the taller flexible frame that suffers the pounding and the column above the contact area are critically increased. It has to be stressed that this is a very common case for adjacent buildings in densely built city centers.

The amplification of the response of a structure that interacts with an adjacent one is mainly observed in the direction of pounding. Thereupon, the response in the perpendicular direction usually remains unaffected unless the contact points of the interacting structures are not symmetric with respect to their plan centers. In this case, due to the asymmetric pounding, torsional movements are induced in the interacting structures [14–16]. The influence of soil–structure interaction on the response of interacting structures has also been studied in recent works [17,18].

Another issue that strongly influences the seismic behavior of reinforced concrete buildings is the distribution of the infills along the height of the structure. Irregular morphology in elevation due to a lack of infills in the first story was the objective of numerous studies. Thus, it is stressed that a soft first story (or open first floor or pilotis) morphology has a crucial role in the seismic response of reinforced concrete buildings [19,20]. Damages in reinforced concrete frame structures during the recent earthquakes indicated that the interaction between masonry infills and the reinforced concrete frame can lead to undesirable effects such as the shear failure of columns, damage to joints and soft-story collapse mechanisms [1,19–21].

In view of the above issues, the influence of masonry infills and pilotis morphology on earthquake-induced pounding between adjacent structures with different total heights was herein investigated using inelastic dynamic analysis. Two types of pounding were studied as follows: pounding between structures that have equal story levels (type A) and pounding where the story levels of the first structure are different from the story levels

of the other one and consequently, the slabs of the floors of each structure hit columns of the other (type B). Furthermore, two cases of masonry infilled frame structures were considered as follows: (a) a frame structure with masonry infills in all stories and (b) a frame structures without infills in the first story (pilotis). Bare frames were also examined in some cases for comparison reasons. Non-linear dynamic analyses and special purpose elements were employed for the needs of this investigation.

## 2. Research Significance

As it became clear from the literature review presented above, the issue of pounding between adjacent structures has been widely studied in the past. However, there are some important aspects that have not been addressed in detail or at all. The present research focuses on some of them.

First of all, one of the objectives of this study is to investigate the response of reinforced concrete frame buildings with pilotis configuration suffering pounding from shorter and stiffer structures. The comparison between such buildings and fully infilled buildings highlights the influence of the open first story on the overall response.

Furthermore, the issue of the directionality of the seismic excitation is examined. In previously presented studies, inelastic dynamic analysis was conducted considering only one direction of ground motion. Obviously, this would be reasonable for symmetric structures. However, pounding abolishes any symmetry. In the present paper, both directions of ground motion were taken into account and the importance of seismic excitation directionality is revealed.

Finally, the present study emphasizes the shear behavior of the external columns suffering the hit at the point of their deformable height (type B). It is demonstrated that the developing shear forces exceed the available strength even for columns with very dense transverse reinforcement. It is apparent that this issue is critical for the overall response and probably constitutes the most serious and dangerous consequence of pounding between adjacent buildings.

## 3. Key Assumptions of the Study

### 3.1. Model Idealization of Pounding Cases between Adjacent Structures

In this work, characteristic cases of structural interaction between two adjoining structures with different total heights were examined. In every case, one multistory fully infilled structure or a structure with an open first story (pilotis) is in contact or very close to one stiffer and shorter structure. Each of the two structures responds dynamically and vibrates independently. In the case of a pre-defined distance ($d_g$) between the interacting buildings, collisions take place only when the relative lateral displacement of the structures exceeds the value of $d_g$. In this study, the influence of the pre-existing $d_g$ size on the pounding effects was also parametrically studied. Two types of structural interaction were identified as the following:

(a) Type A pounding or floor-to-floor pounding (Figure 2a). The floor levels of the interacting structures have equal heights and therefore collisions take place between the slabs of the floors.

(b) Type B pounding or floor-to-column pounding (Figure 2b). In this common case, the heights of the floor levels of the interacting structures are different. Consequently, collisions occur between floors and columns. The slabs of each frame hit a column of the other at the point of its deformable height. The hit that suffers the column of the taller frame by the slab of the upper floor of the shorter structure is usually intense and may prove critical in some cases.

The model idealization of the pounding types is presented in Figure 2. Contact points were considered at the levels of the slabs of the shorter and stiffer structure.

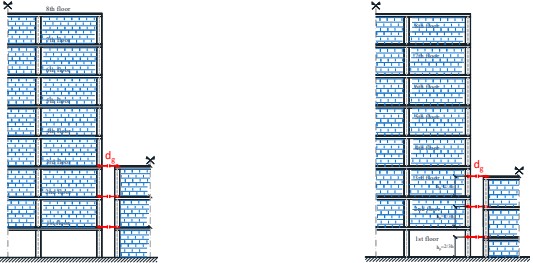

(**a**) Type A or floor-to-floor pounding   (**b**) Type B or floor-to-column pounding

**Figure 2.** Interaction between adjacent structures.

### 3.2. Simulation of Impact Area

The contact areas of the two interacting buildings were simulated using zero-length spring elements that were inactive and became active when at a time step during the seismic analysis, the corresponding nodes of the adjoining frames coincided, meaning that they collided. This approach is consistent with the adopted simulation of the two buildings and appears to be effective for the study of the pounding phenomenon. Further, it is noted that in the case of type B pounding, the flexural and shear deformations, yielding of the reinforcing bars, ductility demands of the columns and other damages to the contact area were considered based on the simulation of the column element.

The response of the contact element comprises three phases. The first phase is the condition under which the two interacting structures move away from one another; it is represented by the negative direction of the *x*-axis while the contact element remains inactive. Further, if there is an initial distance $d_g$ between the structures, there is a phase during which the two structures move in such a way that they are getting closer, but the displacements are small and the structures continue to vibrate independently. This phase is represented by the first part of the positive *x*-axis being equal to $d_g$ and it is noted that in this phase, the contact element remains inactive. Finally, the third phase is the case that the buildings move in such a way that they bridge the pre-existing distance between them or they are in contact from the beginning and the contact element reacts as a spring (Figure 3). The value of the stiffness of this spring element is highly uncertain because the materials that suffer the impact present uncertain properties and the geometries of the impact surfaces are also unknown; nevertheless, from the literature, the system behavior is not sensitive to the stiffness variation of the spring element [4,16,22,23]. Furthermore, it is noted that the influence of the damping values of the spring element on the response of the two-building system has also been studied and proven to be negligible [16].

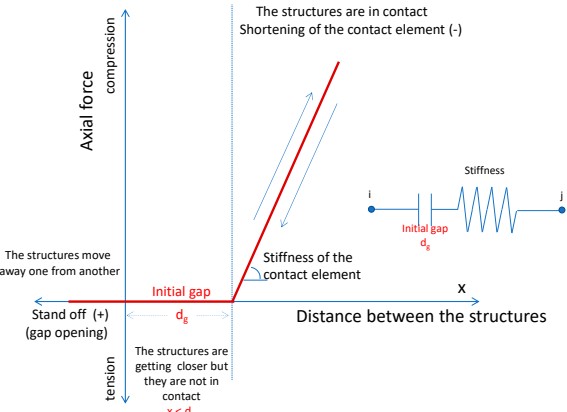

**Figure 3.** Contact element. The response of contact elements includes three parts: the structures move away one from another, represented by the negative *x*-axis; the structures get closer but are not in contact, represented by the first part of the positive *x*-axis equal to $d_g$; and the structures are in contact and the contact element reacts as a spring with almost infinite stiffness, represented by the positive *x*-axis for $x \geq d_g$.

### 3.3. Simulation of Interacting Multistory Structures

The 8-story reinforced concrete frame designed according to Eurocodes 2 and 8 interacts with adjacent shorter and stiffer buildings. Its concrete is of class C20/25 ($f_{ck}$ = 20 MPa) and the reinforcing steel bars B500C ($f_{yk}$ = 500 MPa), according to Greek standards. The adopted design factors were the following: ductility class medium (DCM), behavior factor q = 3.9 and design ground acceleration $a_g$ = 0.3 g.

The examined cases included the interaction of the 8-story frame with single-story buildings and with 4-story buildings. In particular, three cases of the short structures were considered: a 4-story reinforced concrete frame structure (fundamental period $T_1$ = 0.311 s), a single-story masonry structure ($T_1$ = 0.072 s) and a very stiff non-self-vibrating one represented by rigid nodes connected with the impact springs at the contact points of the structure that suffered the hit. All studied interaction cases included pounding cases type A (floor-to-floor) and type B (floor-to-column).

This study focuses on the influence of the open first story (pilotis) on the pounding phenomenon. Therefore, all examined two-building poundings were studied considering two cases: the first case of a fully infilled 8-story frame and the second case of an infilled 8-story with an open first story (pilotis). The fundamental periods of the frames were 0.602 s and 1.022 s, respectively.

The flexural and shear stiffness of members was taken as equal to one-half of the corresponding stiffness of the uncracked members. The columns were considered fixed at the base. Concerning the inelastic behavior of the concrete members, it was assumed that plastic deformations were concentrated (i) at the critical sections, i.e., at the ends of the structural components and (ii) at the contact point in cases of type B pounding. Plastic hinges were modeled by moment–rotation diagrams and the moment–axial force interaction in columns was taken into account by the appropriate interaction surface.

For the simulation of the infills, the equivalent strut model was employed (Figure 4a). Two different types of elements have been examined in the literature [24]. The first was a truss element with zero strength in tension and a bilinear response in compression. In the analysis process, if the truss element reached its ultimate strength, it was assumed that the total infilled frame strength became equal to the value of the strength of the bare frame at this value of the loading. The other element applied in the current work takes into account the response of the infill masonry more accurately. It includes a degrading branch and residual strength (Figure 4b). More specifically, the axial behavior of the diagonal struts is defined by five characteristic points (S1 to S5) with each one corresponding to a discrete state of the strut: S1 is the first yield point, S2 is an intermediate point between S1 and S3 where remarkable stiffness reduction occurs, S3 designates the maximum strength of the strut, while S4 and S5 are the starting points of the degrading and residual strength branches, respectively. It was important that both elements exhibited an axial response only and not flexural properties.

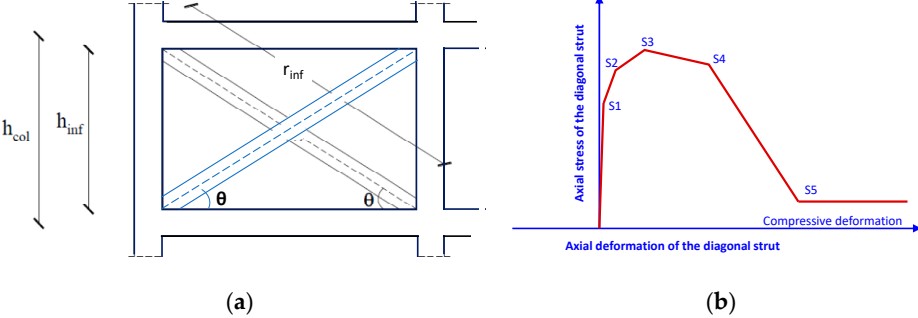

| (a) | (b) |

**Figure 4.** Diagonal strut model. (**a**) The infill masonry is simulated using two equivalent diagonal struts. (**b**) Each strut can only carry compressional axial loading.

Furthermore, an important issue was the determination of the response characteristics of the strut element in order to accurately represent the properties of the infill masonry. In

this study, the actual property values of the considering infills were assumed considering experimental values by Karayannis et al. [24] and Kakaletsis and Karayannis [25]. The effective width of the diagonal strut was calculated according to the following relationships by FEMA 273 [26] and FEMA 306 [27] that are mainly based on Mainstone [28]:

$$w = 0.175(\lambda_1 h_{col})^{-0.4} r_{inf} \tag{1}$$

$$\lambda_1 = \left[ \frac{E_{me} t_{inf} sin2\theta}{4EIh_{inf}} \right]^{1/4} \tag{2}$$

where $E_{me}$ is the modulus of elasticity of masonry, $t_{inf}$ is the thickness of the infill masonry, $EI$ is the flexural stiffness of columns, $h_{col}$ is the story height, $h_{inf}$ is the clear story height and $r_{inf}$ and $\theta$ were geometrically calculated (see Figure 4).

*3.4. Examined Interaction Cases*

Based on contemporary design philosophy, the concept of strong columns and weak beams is usually applied and therefore flexible RC frame structures were designed and constructed. Further, as reported repeatedly [11–17] in pounding cases between two adjoining structures, more vulnerable to pounding damage is the taller and more flexural structure that usually suffers the hit from the slabs of the shorter and stiffer one during the seismic excitation. On the other hand, the frequent adoption of an open first story configuration (pilotis) for certain architectural and functionality reasons is common practice although it is widely acceptable that pilotis buildings frequently suffer severe damage during earthquakes.

Thereupon, in this investigation, the influence of the open first story morphology on the structural pounding was evaluated based on the behavior and the local response characteristics of critical columns of an 8-story frame structure (Figure 5) designed according to Eurocodes 2 [29] and 8 [6]. Henceforth, the considered pounding cases involved the interaction between an 8-story RC frame with (a) a single-story structure (Figure 6) and (b) a 4-story structure (Figure 7). Each case was studied for pounding case A and pounding case B as described in paragraph 2.1. Furthermore, each time, the 8-story frame was considered (a) as a fully infilled frame and (b) as an infilled frame with an open first story. It is noted that three cases of short structures were taken into account: a frame structure, a stiff structure and a very stiff non-self-vibrating one represented by rigid nodes. Finally, all cases were examined considering that the two interacting structures were in contact from the beginning.

**Figure 5.** Structural characteristics of an 8-story RC frame.

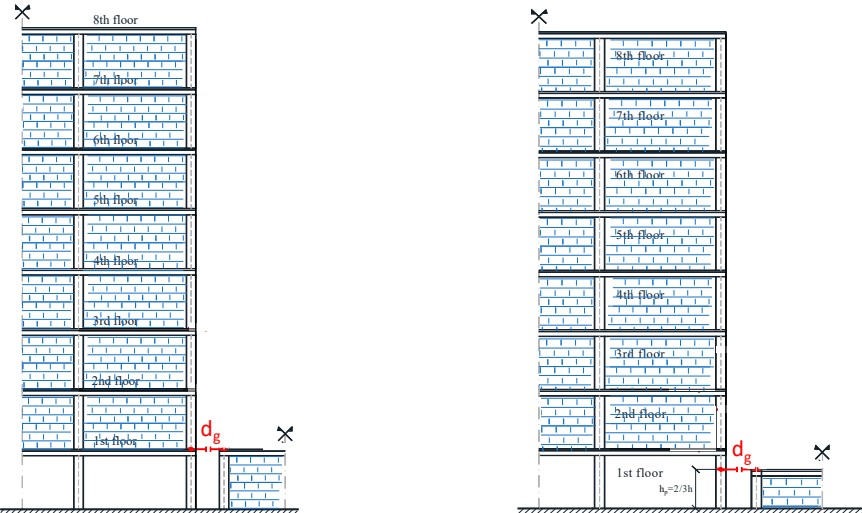

(**a**) Type A or floor-to-floor pounding    (**b**) Type B or floor-to-column pounding

**Figure 6.** Interaction between the 8-story pilotis frame and a single-story structure.

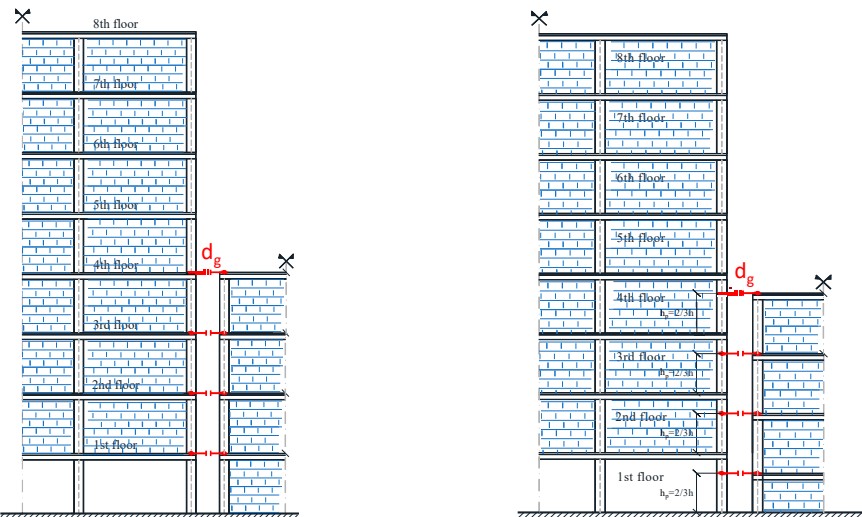

(**a**) Type A or floor-to-floor pounding    (**b**) Type B or floor-to-column pounding

**Figure 7.** Interaction between an 8-story pilotis frame and a 4-story structure.

### 3.5. Considered Seismic Excitations

Each examined pounding case was subjected to two different natural seismic excitations: (a) The El Centro 1940 earthquake (duration first 16 s) and a maximum ground acceleration $\alpha_{max}$ = 0.318 g and (b) the Loma Prieta 1989 earthquake (duration first 16 s) and a maximum acceleration $\alpha_{max}$ = 0.29 g. The time interval of 16 s always included the peak response of the structures. It is noted that the maximum accelerations of both seismograms were almost equal to the designed seismic acceleration of the 8-story frame structure ($a_g$ = 0.30 g). The accelerograms and the relevant response spectra are illustrated in Figures 8 and 9, respectively.

Moreover, the plus and minus direction of the seismic excitation in relation to the 8-story frame proved to be of utmost importance for the evaluation of the developing capacity demands due to the pounding. This was mainly attributable to the asymmetry of the examined two-structured pounding pairs and it is emphasized that in these cases, it is vital that the plus and minus directions of the seismic excitation have to be separately examined.

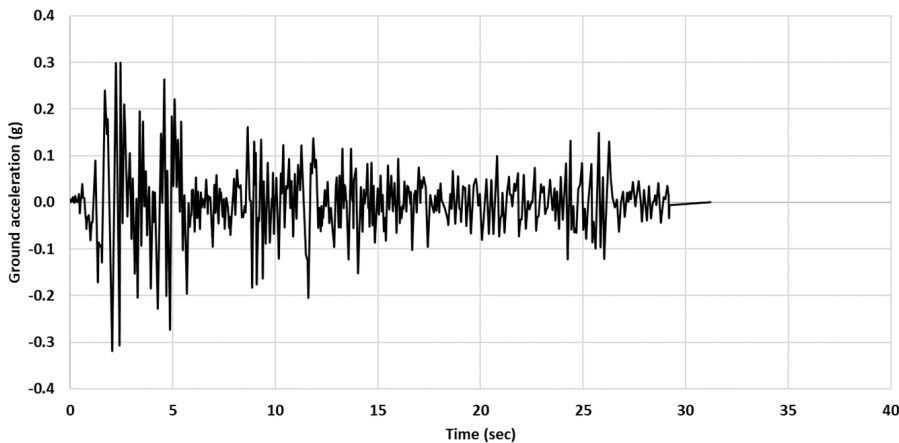

(**a**) El Centro 1940

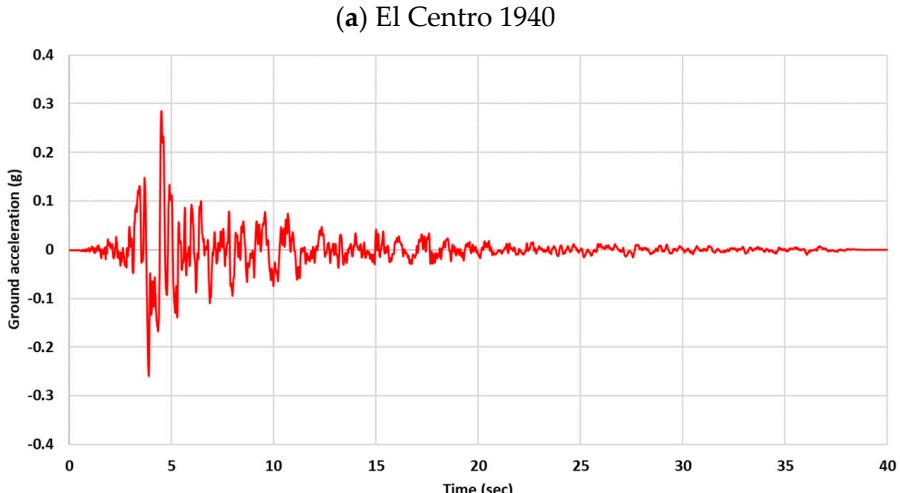

(**b**) Loma Prieta 1989

**Figure 8.** Accelerogram of seismic excitations.

**Figure 9.** Pseudo-acceleration response spectra of seismic excitations.

## 4. Results

*4.1. Pounding at the First Floor (Pilotis Floor) Level*

Interaction cases between the 8-story frame and a single-story structure at the level of the first floor were examined considering pounding (Figure 6): (a) at the floor level (type A

pounding or floor-to-floor pounding) and (b) at a point within the deformable height of the external column of the 8-story frame (type B pounding or floor-to-column pounding). Each pounding case was subjected to two natural seismic excitations (El Centro 1940 and Loma Prieta 1989) and each excitation was applied twice (+ direction and − direction).

### 4.1.1. Pounding at the Floor Level (Type A Pounding)

The comparative results of the floor-to-floor pounding between the 8-story frame and a single-story masonry structure considering the 8-story frame was fully infilled or infilled with an open first story (pilotis) are presented in Figure 10. Time histories of the 1st floor displacements for the cases of pounding of a (i) fully infilled frame with a single-story masonry structure (red line) and (ii) pilotis frame with a single-story masonry structure are presented in Figure 10. In the same figure, the maximum interstory drifts of the 8-story frame for these pounding cases are also presented. It was observed that the pilotis configuration increased the maximum first floor interstory drift and caused significant permanent deformations due to the pounding. On the contrary, interstory drifts of the upper stories of the frames did not essentially differ. Moreover, considerable differences between the two directions of ground motion were observed.

Moreover, the results of floor-to-floor pounding between the 8-story frame and a single-story very stiff structure considering the 8-story frame was (i) fully infilled or (ii) infilled with an open first story (pilotis) are presented in Figure 11. In these cases, the pilotis configuration also significantly increased the maximum first floor interstory drift due to the pounding. At the same time, the role of the ground motion directionality was even more pronounced.

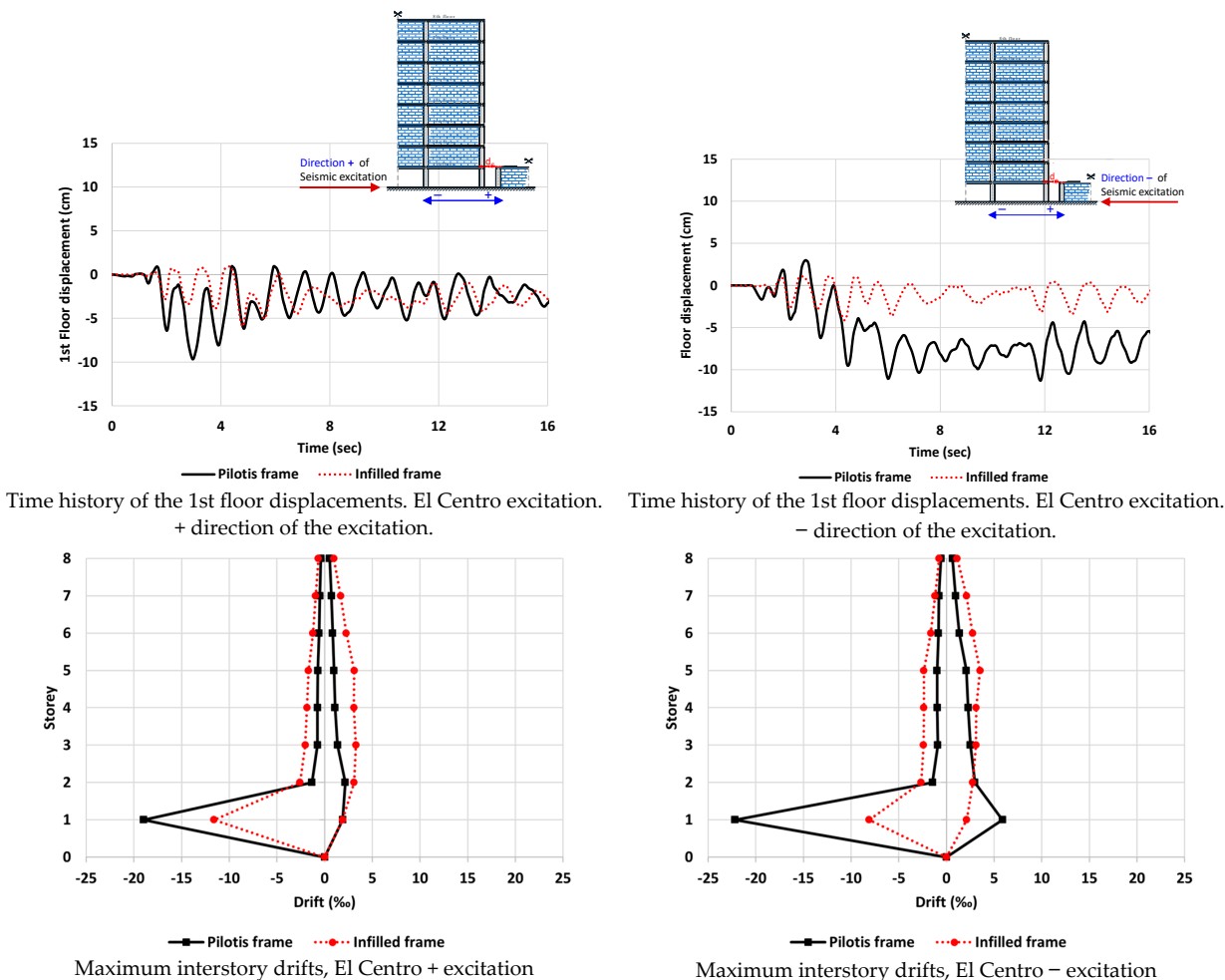

**Figure 10.** *Cont.*

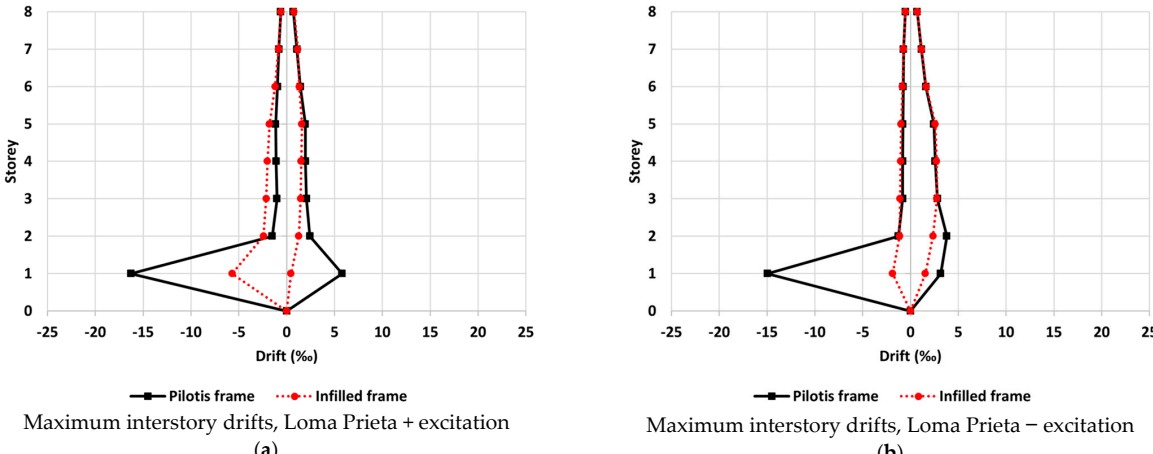

Maximum interstory drifts, Loma Prieta + excitation
(**a**)

Maximum interstory drifts, Loma Prieta − excitation
(**b**)

**Figure 10.** Floor-to-floor pounding between the 8-story pilotis and a single-story masonry structure. Time history of the 1st floor displacements and maximum interstory drifts of the 8-story frame. It was observed that pilotis configuration significantly increased the maximum first floor interstory drift due to the pounding. (**a**) Time history of 1st floor displacements and story drifts of the 8-story frame in the + direction of seismic excitation. (**b**) Time history of 1st floor displacements and story drifts of the 8-story frame in the − direction of seismic excitation.

It is worth noting that type A pounding may cause significant damage to exterior beam–column joints which are the most vulnerable structural components of modern buildings [30–33].

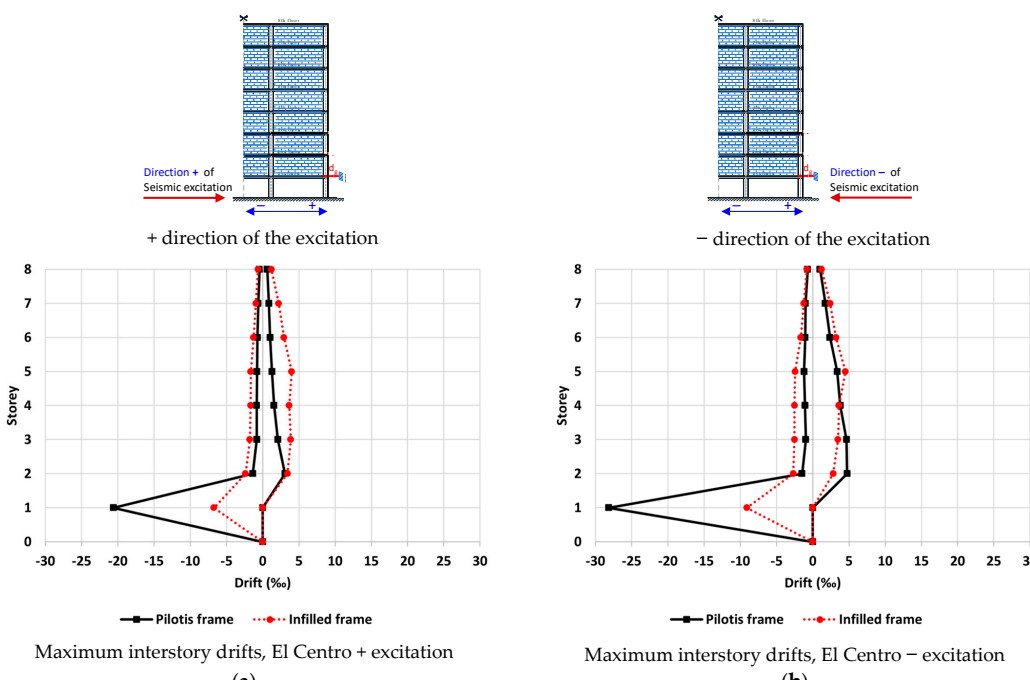

Maximum interstory drifts, El Centro + excitation
(**a**)

Maximum interstory drifts, El Centro − excitation
(**b**)

**Figure 11.** Floor-to-floor pounding between the 8-story pilotis frame and a very stiff structure. Time history of the 1st floor displacements and maximum interstory drifts of the 8-story frame. It was observed that pilotis configuration significantly increased the maximum first floor interstory drift due to the pounding. (**a**) Maximum interstory story drifts of the 8-story frame in the + direction of seismic excitation. (**b**) Maximum interstory story drifts of the 8-story frame in the − direction of seismic excitation.

4.1.2. Floor-to-Column Pounding (Type B Pounding)

The comparative results of floor-to-column pounding between the 8-story frame and a single-story masonry structure with a floor height lower than the 1st story height of the 8-

story frame considering the 8-story frame was (i) fully infilled or (ii) infilled with open first story (pilotis) are presented in Figure 12. Time histories of the 1st floor displacements for the cases of pounding of a (i) fully infilled frame with single-story masonry structure (red line) and (ii) pilotis frame with single-story masonry structure are presented in Figure 12a,b, respectively. In the same figure, the maximum interstory drifts of the 8-story frame for both pounding cases are also presented. It was observed that the pilotis configuration increased the maximum first floor interstory drift due to the pounding. However, the drift amplification was, in general, less significant in comparison with type A pounding.

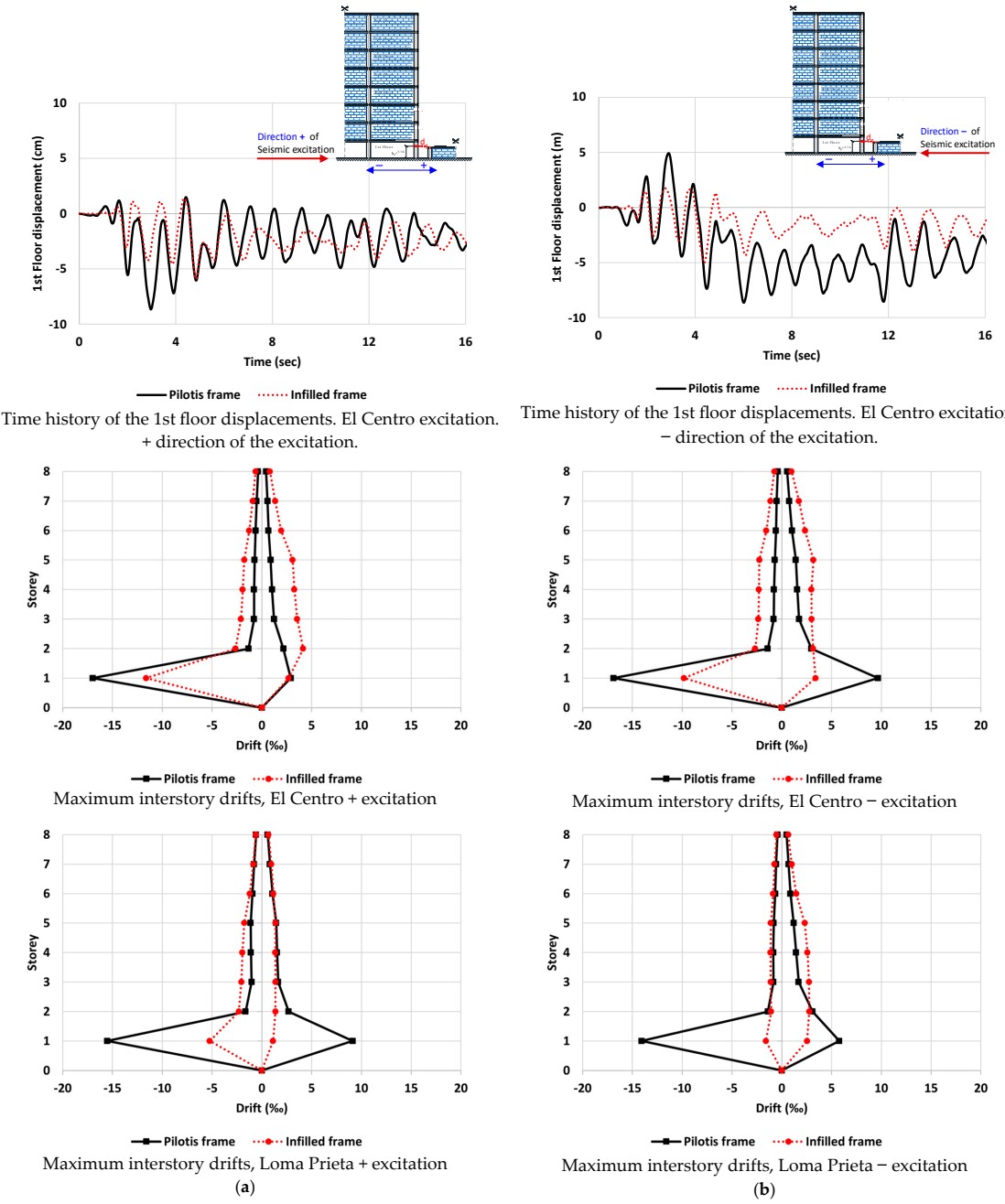

**Figure 12.** Floor-to-column pounding between the 8-story pilotis and a single-story masonry structure. Time history of the 1st floor displacements and maximum interstory drifts of the 8-story frame. It was concluded that the pilotis configuration significantly increased the maximum first floor story drift due to the pounding. (**a**) Time history of 1st floor displacements and story drifts of the 8-story frame in the + direction of seismic excitation. (**b**) Time history of 1st floor displacements and story drifts of the 8-story frame in the − direction of seismic excitation.

Moreover, the maximum interstory drifts of floor-to-column pounding between the 8-story frame and a single-story very stiff structure considering the 8-story frame was (i) fully infilled or (ii) infilled with an open first story (pilotis) are presented in Figure 13. In these cases, pilotis configuration also significantly increased the maximum first floor interstory drift due to the pounding.

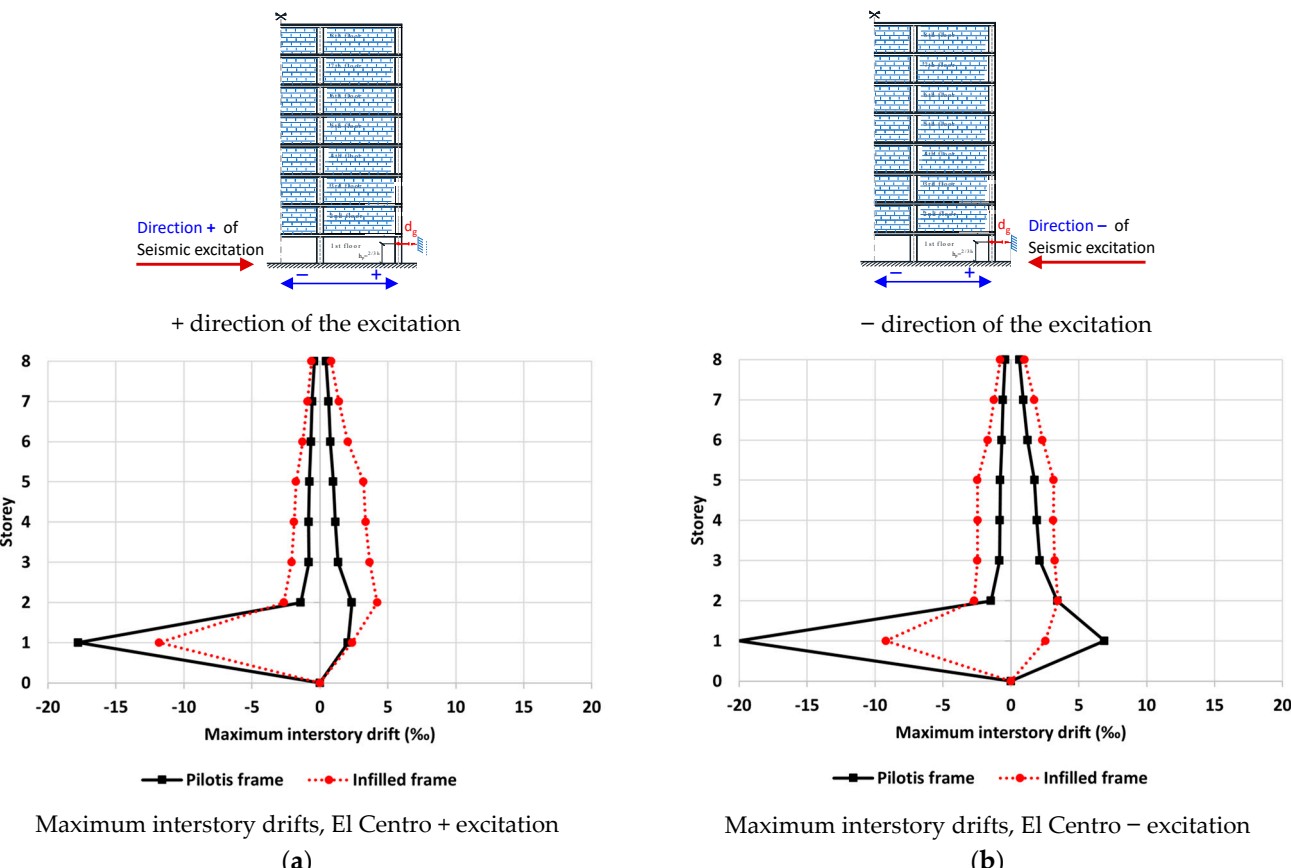

Maximum interstory drifts, El Centro + excitation

(**a**)

Maximum interstory drifts, El Centro − excitation

(**b**)

**Figure 13.** Pounding between the 8-story pilotis frame and a very stiff structure. Time history of the 1st floor displacements and maximum interstory drifts of the 8-story frame. It was concluded that pilotis configuration significantly increased the maximum first floor story drift due to the pounding. (**a**) Maximum interstory drifts of the 8-story frame in the + direction of seismic excitation. (**b**) Maximum interstory drifts of the 8-story frame in the − direction of seismic excitation.

### 4.1.3. Pounding Forces

The developing pounding forces during the interaction of the 8-story frame with the single-story structures for floor-to-column pounding are presented in Figures 14 and 15 for the pounding cases of an (i) 8-story frame with single-story masonry structure and (ii) 8-story frame with single-story very stiff structure, respectively.

During the time intervals that the two adjacent structures were not in contact, pounding forces were equal to zero. On the other hand, when the two adjacent structures were in contact, pounding forces were below zero. The negative sign designates that the forces were compressive. From Figures 14 and 15, it is clearly observed that the pounding forces significantly increased in the cases of an infill 8-story frame with an open first story (pilotis configuration). However, in most cases, the development of pounding forces in the pilotis frame was limited to the first few seconds of the excitation. This is due to the fact that significant permanent deformations occurred and the structures moved away from each other.

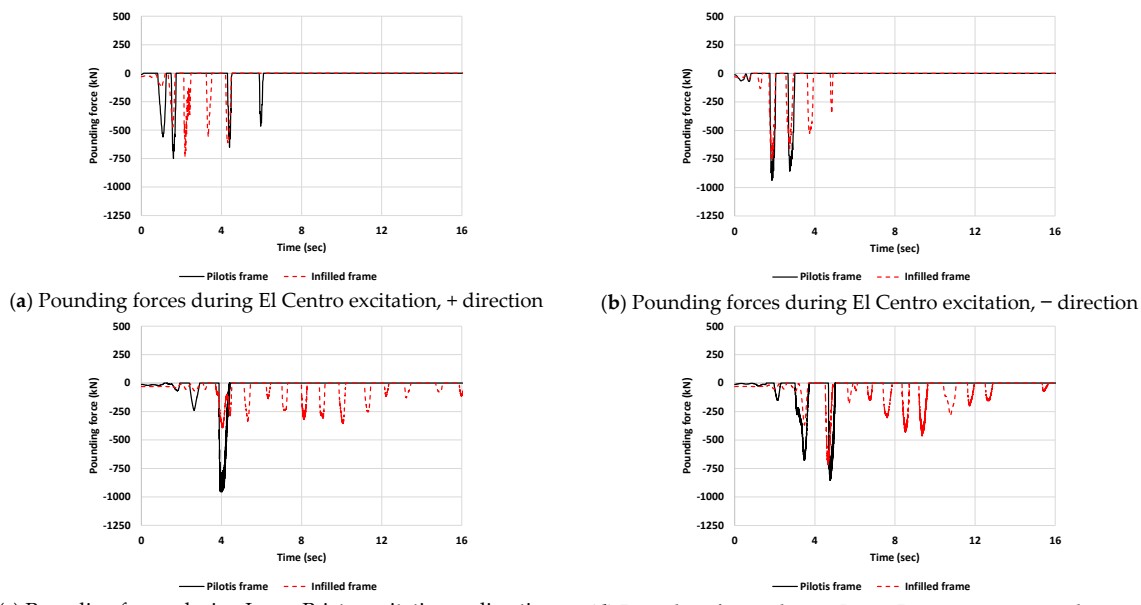

(**a**) Pounding forces during El Centro excitation, + direction

(**b**) Pounding forces during El Centro excitation, − direction

(**c**) Pounding forces during Loma Prieta excitation, + direction

(**d**) Pounding forces during Loma Prieta excitation, − direction

**Figure 14.** Floor-to-column pounding. Developing forces during pounding between the 8-story frame and a single-story masonry structure. It was observed that pounding forces increased in the case of the pilotis frame.

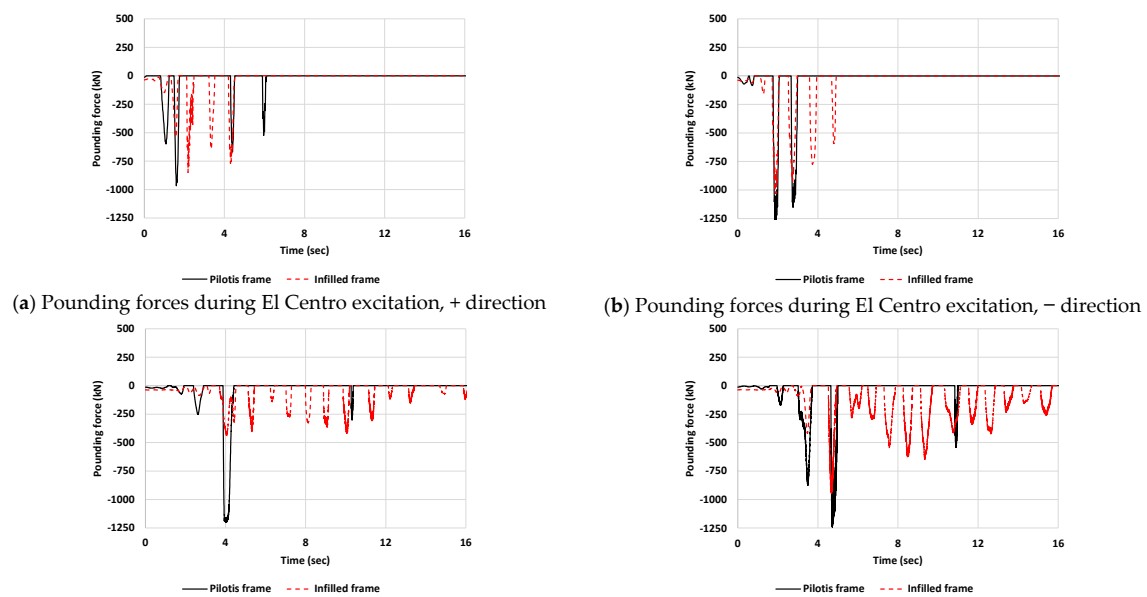

(**a**) Pounding forces during El Centro excitation, + direction

(**b**) Pounding forces during El Centro excitation, − direction

(**c**) Pounding forces during Loma Prieta excitation, + direction

(**d**) Pounding forces during Loma Prieta excitation, − direction

**Figure 15.** Floor-to-column pounding. Developing forces during pounding between the 8-story pilotis frame and a single-story very stiff structure. It was observed that pounding forces increased in the case of the pilotis frame.

### 4.2. Pounding between the 8-Story Frame and a 4-Story Structure

The interaction cases between the 8-story frame and a 4-story structure were also examined as follows: (a) considering pounding at the floor levels (type A pounding or floor-to-floor pounding) and (b) considering pounding cases where the hit took place at a point within the deformable height of the external columns of the 8-story frame (type B pounding or floor-to-column pounding). Each pounding case was subjected to two natural seismic excitations (El Centro 1940 and Loma Prieta 1989) and each excitation was applied twice (+ direction and − direction).

### 4.2.1. Maximum Interstory Drifts

The comparative results of the floor-to-column pounding between the 8-story frame and the 4-story structure with the floor levels lower than the corresponding levels of the 8-story frame are presented in Figure 16. This study included two cases: (i) a fully infilled 8-story frame and (ii) an infilled structure with an open first story (pilotis). The maximum interstory drifts of the 8-story frame for the studied cases are presented in Figure 16. It was observed that the pilotis configuration significantly increased the maximum first floor interstory drift due to the pounding, while the upper story drifts were not remarkably affected. The importance of ground motion directionality is highlighted once again.

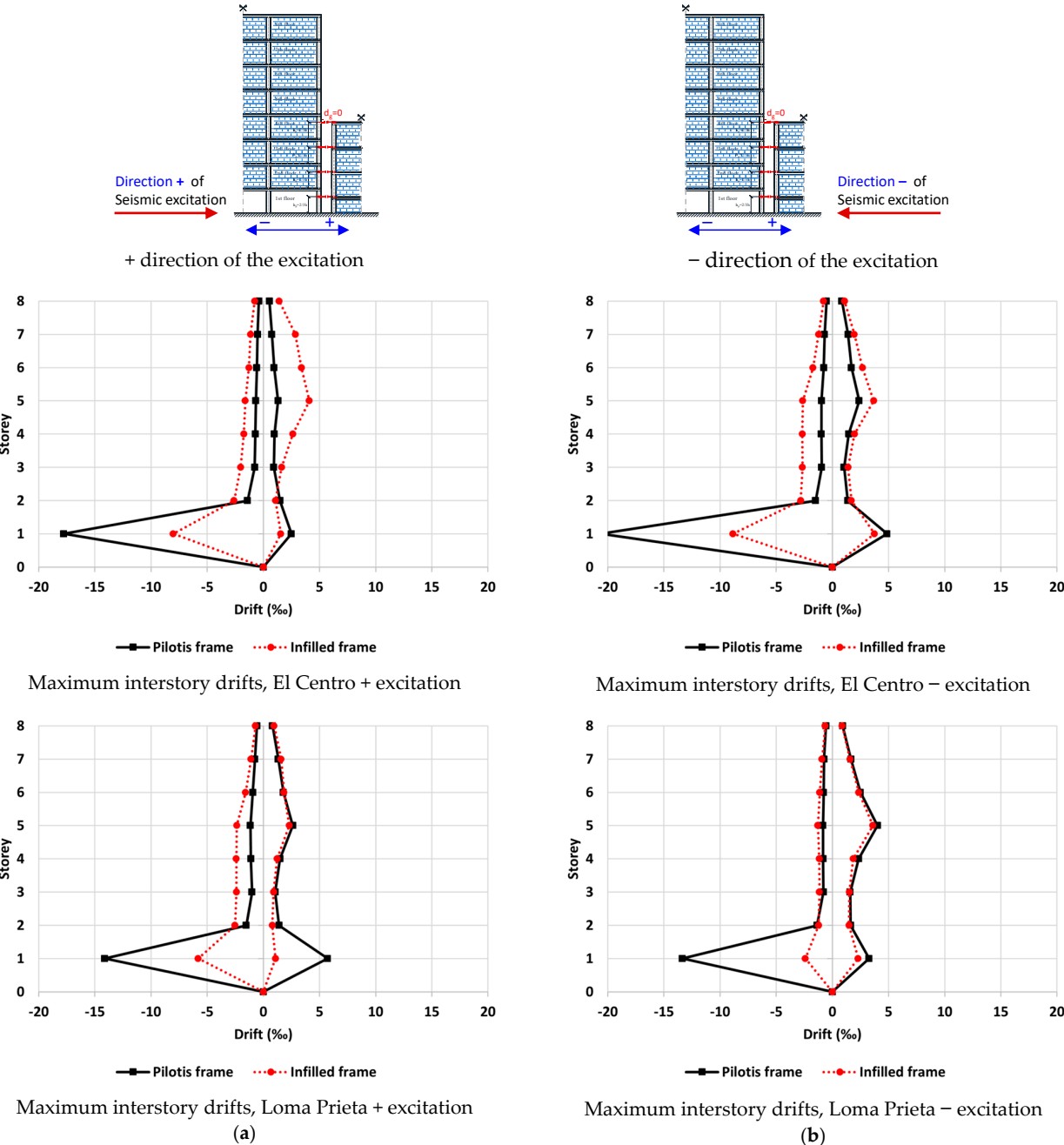

**Figure 16.** Floor-to-column pounding between the 8-story pilotis frame and a 4-story infilled structure. Maximum interstory drifts of the 8-story frame. (**a**) Maximum interstory drifts of the 8-story frame in the + direction of seismic excitations. (**b**) Maximum interstory drifts of the 8-story frame in the − direction of seismic excitations.

4.2.2. Shear Forces in the Column That Suffered the Hit

An important issue was the developing shear forces of the columns of the 8-story frame that suffered the hit from the corresponding floor slabs of the adjacent 4-story structure. The developing shear forces from the El Centro excitation in the − and + directions of the external columns of the 4th and the 3rd story of the 8-story frame that suffered the pounding and a comparison with the available shear strength of the columns are presented in Figure 17. In particular, the shear forces of the infilled frame with an open first story (pilotis) are illustrated in Figure 17a,c and the shear forces of the fully infilled frame, in Figure 17b,d. The shear strength of the columns were estimated based on Eurocode 2 [6].

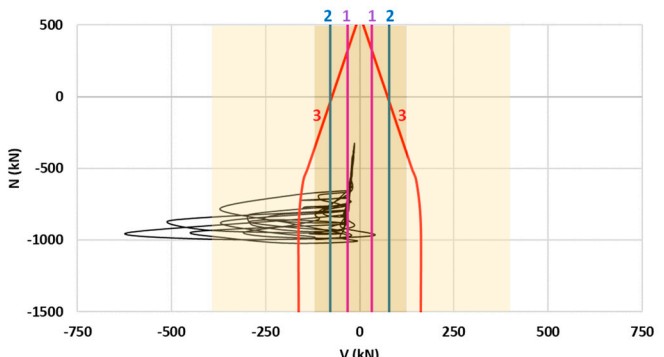

(**a**) 8-story frame pilotis, shear forces of the column of the 4th floor that suffered the hit

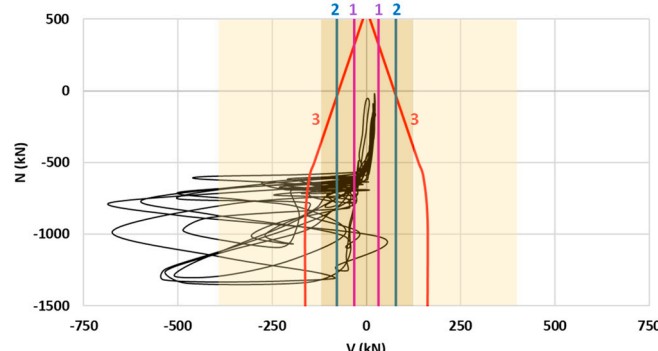

(**b**) 8-story infilled frame, shear forces of the column of the 4th floor that suffered the hit

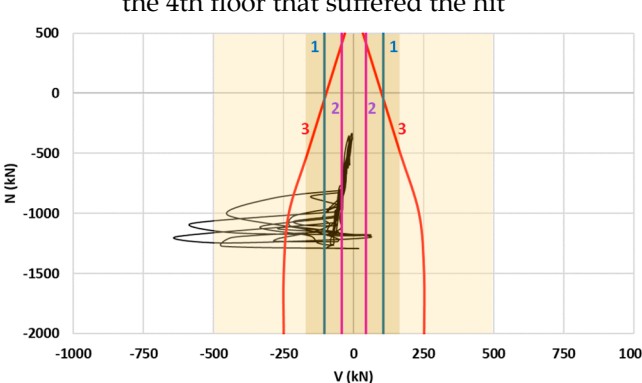

(**c**) 8-story frame pilotis, shear forces of the column of the 3rd floor that suffered the hit

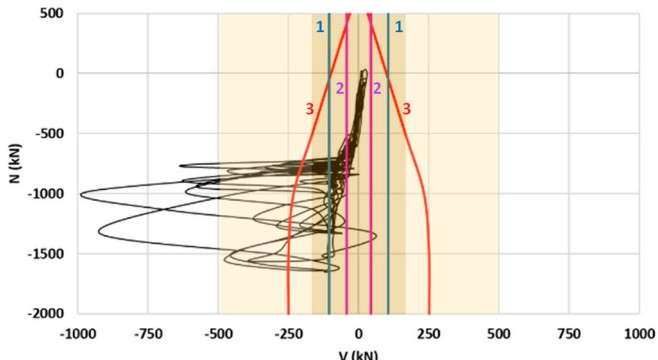

(**d**) 8-story infilled frame, shear forces of the column of the 3rd floor that suffered the hit

Black lines: shear force (V)—axial force (N) history
1: shear strength due to stirrups according to the EC2 (low limit, crack angle θ = 45° to EC8) for the non-critical region
2: shear strength due to stirrups according to the EC2 (upper limit, crack angle θ = 22° to EC8) for the non-critical region
3: shear strength due to concrete (EC2) without stirrups
**Dark yellow area**: shear strength if stirrups for critical regions (EC8) are applied (low limit, crack angle θ = 45° to EC8)
Light yellow area: shear strength if stirrups for critical regions (EC8) are applied (upper limit, crack angle θ = 22° to EC8)

**Figure 17.** Floor-to-column pounding between the 8-story pilotis frame and a 4-story infilled frame for the El Centro excitation in the − and + directions. Developing shear forces of the external column of the 4th story and 3rd story that suffered the pounding (**a**) and (**c**) pilotis (**b**) and (**d**) infilled. Developing shears due to the hit within the deformable height of the column exceeded the shear strength of the column even if the whole height of the column was considered as critical according to EC8.

Since the shear response of these columns is critical for the whole structure and for a thorough understanding of the issue, the shear strength was calculated based on three alternative considerations:

- stirrups according to the EC2 [6] and EC8 [29], low limit (crack angle θ = 45°) and upper limit (crack angle θ = 22°)
- shear strength due to concrete (EC2) without stirrups
- shear strength considering stirrups as if the whole height of the column was the critical region (EC8—low limit crack angle θ = 45° and upper limit crack angle θ = 22°).

As can be observed in Figure 17, the developing shears due to the hit within the deformable height of the column exceeded the shear strength of the column many times during the seismic excitation even if the whole height of the column was considered as critical according to the EC8. Further, it is noted that the shear strength of the reinforced concrete elements according to the EC2 depended on the angle θ of the developing cracks with the axis of the element. Thereupon, the uncertain character of the assumed strength was stressed especially in the case that the angle θ was equal to 22°. Therefore, as shown in Figure 17, it can be concluded that in the examined cases, the columns that suffered the hit experienced shear failure.

## 5. Discussion

The results presented in the previous section aim to highlight the importance of three specific aspects that are relevant to pounding:

- the influence of the pilotis in comparison to fully infilled frames;
- the influence of seismic excitation directionality;
- the shear behavior of columns that suffer the hit.

However, it would also be useful to compare the seismic response of buildings with and without pounding. For this purpose, maximum interstory drifts under El Centro excitation of floor-to-column pounding between the 8-story frame and a single-story very stiff structure are presented in Figures 18 and 19 for the pilotis and fully infilled frame, respectively. As expected, pounding remarkably increased interstory drifts especially for the first story which suffered the hit. Similar results were obtained for the other pounding cases too; however, for the sake of brevity, they are not presented here. It is worth noting that the whole analysis process conducted in the framework of the present paper provides a useful tool for engineering practitioners. In particular, it could be applied to new as well as existing buildings for the assessment of the response under pounding from an adjacent structure and the verification of the gap adequacy. It should be clarified that the determination of the required gap size in order to avoid collision is beyond the objective of the present work. This issue has been widely investigated in the past and relevant equations have been proposed [34,35]. It should also be clarified that the results presented above refer to buildings in contact ($d_g = 0$). The results for separation distances $d_g > 0$ will be presented in a forthcoming paper.

Since this analytical process is based on an inelastic dynamic analysis, it has all the well-recognized shortcomings of the latter: computational cost, selection and scaling of accelerograms, reliability of nonlinear models, etc. Concerning nonlinear modeling, additional issues arise due to the need to simulate the masonry infills as well as the impact area. Thus, finite element models should be selected and verified carefully [36]. In the present paper, well-established analytical models, verified and widely used in similar studies, were adopted (see Section 3).

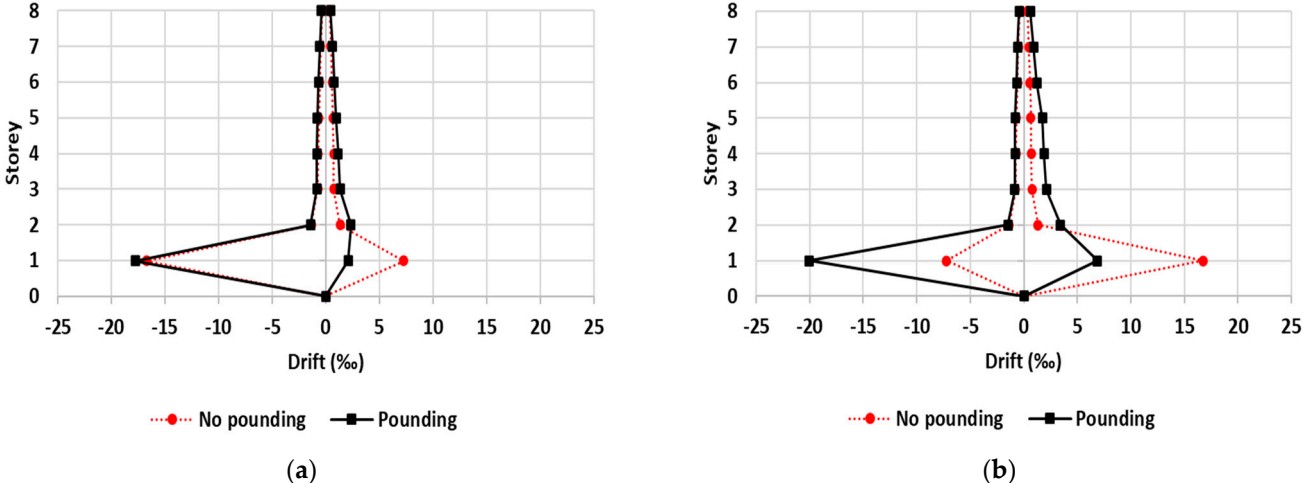

**Figure 18.** Floor-to-column pounding between the 8-story pilotis frame and a single-story very stiff structure during El Centro excitation. Maximum interstory drifts of the 8-story frame. (**a**) Maximum interstory drifts of the 8-story frame in the + direction of El Centro excitation. (**b**) Maximum interstory drifts of the 8-story frame in the − direction of El Centro excitation.

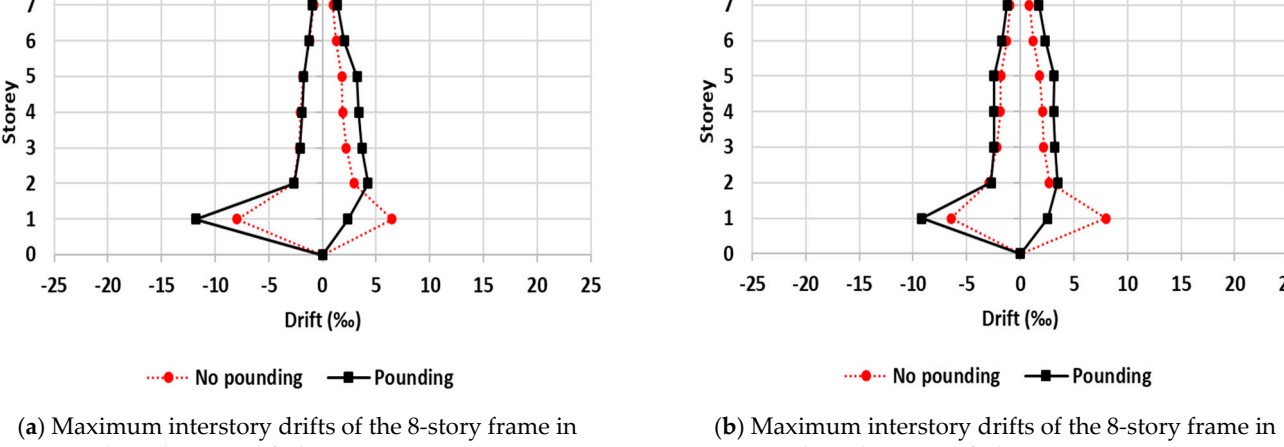

(**a**) Maximum interstory drifts of the 8-story frame in the + direction of El Centro excitation.

(**b**) Maximum interstory drifts of the 8-story frame in the − direction of El Centro excitation.

**Figure 19.** Floor-to-column pounding between the 8-story fully infilled frame and a single-story very stiff structure during El Centro excitation. Maximum interstory drifts of the 8-story frame.

## 6. Conclusions

In the present paper, an extensive analytical study of the seismic interaction between adjacent structures was conducted and presented. In particular, two nearly identical 8-story reinforced concrete planar frames were designed according to Eurocodes 2 and 8. The only difference between the frames is that the first one is fully infilled while the other has an open first story (pilotis). It was considered that the examined frames are in contact with three different types of structures: a 4-story reinforced concrete frame, a single-story masonry structure and a very stiff non-self-vibrating structure represented by rigid nodes at the contact points. Two different types of pounding were examined: floor-to-floor (type A) and floor-to-column pounding (type B). The structures were analyzed by means of inelastic dynamic analysis for two recorded accelerograms with peak ground acceleration almost equal to the design ground acceleration (≈0.30 g). The accelerograms were applied along both directions and representative response quantities (floor displacements, interstory drifts, shear forces) were calculated.

This whole study aims to investigate crucial aspects of the pounding phenomenon. More specifically, it focuses on the influence of pilotis configuration in comparison with

fully infilled frames, on the influence of seismic excitation directionality and on the shear behavior of columns that suffer the hit at the point of their deformable height. The main derivations of this study are as follows:

- The findings of previous research which indicated that pounding may cause significant amplification of the seismic response are confirmed once again.
- It is inferred that pilotis configuration increases the developing pounding forces as well as the deformations of the structural elements and consequently increases their capacity demands.
- For the first time, it is stressed that pounding between structures must be examined in both directions (plus and minus) of each seismic excitation.
- It was deduced that the developing shear forces on the columns that suffered the hit in the case of type B pounding exceeded the shear strength of the column even if detailing for critical regions according to the Eurocode 8 was applied.

Finally, it is worth noting that the generalization of the above conclusions requires further investigations, comprising applications to a large variety of structures and using an adequately high number of earthquake ground motions.

**Author Contributions:** Conceptualization, C.G.K. and G.E.M.; methodology, C.G.K.; software, G.E.M.; validation, C.G.K. and G.E.M.; investigation, C.G.K. and G.E.M.; data curation, G.E.M.; writing—original draft preparation, C.G.K.; writing—review and editing C.G.K.; visualization, C.G.K. and G.E.M. All authors have read and agreed to the published version of the manuscript.

**Funding:** This research received no external funding.

**Data Availability Statement:** Data sharing not applicable.

**Conflicts of Interest:** The authors declare no conflict of interest.

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
