# Peer review of "Seismic Interaction between Multistory Pilotis RC Frames and Shorter Structures with Different Story Levels—Floor-to-Column Pounding"

_2673-4109, doi:10.3390/civileng4020036_

Round 1
Reviewer 1 Report
There are some smaller problems with page formatting, some grammar correction (the necessary corrections were indicated in the attached document), etc.

Author Response
Reviewer 1
There are some smaller problems with page formatting, some grammar correction (the necessary corrections were indicated in the attached document), etc.
We would like to thank the reviewer for his efforts. All corrections indicated are made in the revised manuscript and really contributed to the improvement of the paper.
Reviewer 2 Report
In this article, characteristic cases of structural interaction between two adjoining struc- 138 tures with different total heights are examined. Two states including floor-to-floor floor-to-column were considered analytically. The subject matter is interesting and valuable. The following comments may help the authors to improve their presentation:
1- The novelty of the work must be clarified.
2- Since the article is an analytical study, a verification should be added. The article “Evaluation and Verification of Finite Element Analytical Models in Reinforce Concrete Members” may be useful.
3- Please define each of the states S1 to S5 that have been considered in stress/deformation graph for the strut model (see Fig. 4).
4- Adding two new sections would be helpful: research significance (after introduction), and discussion (before conclusion). In the "Research Significance" section, explain the novelty and research significance in one or two paragraphs. In the "Discussion" section, tell us about the pros and cons, benefits and also limitations of the proposed
5- Please provide figures of full analytical frame within the article. Moreover, please add full details about these two frames.
6- It is not clear how the authors determined initial gap between two structures. In this regard, the article “Prediction of critical distance between two MDOF systems subjected to seismic excitation in terms of artificial neural networks” introduce a technique to estimate the gap. Please consider such references in your work.
7- Please clarify how the parameters such as hcol, hinf, rinf, were calculated in this study.
8- The figures presented in section 3, needs to be described and interpreted more.
9- It is not clear what the black lines are in Fig. 15.
10- Conclusion should be extended.
Author Response
Reviewer 2
In this article, characteristic cases of structural interaction between two adjoining structures with different total heights are examined. Two states including floor-to-floor floor-to-column were considered analytically. The subject matter is interesting and valuable. The following comments may help the authors to improve their presentation:
First of all, we would like to thank the reviewer for his constructive and helpful comments which are incorporated in the revised manuscript. Here are some clarifications concerning each one of them:
1- The novelty of the work must be clarified.
Following the kind suggestion of the reviewer a quite new paragraph has been added in the revised manuscript. Thus, novelty of the work is summarized and clarified in “Research Significance” section of the revised manuscript.
2- Since the article is an analytical study, a verification should be added. The article “Evaluation and Verification of Finite Element Analytical Models in Reinforced Concrete Members” may be useful.
Obviously, the verification of the analytical models adopted is a very important issue. For the present paper, well-established analytical models, verified and widely used in similar studies are adopted. The known and well-established program SAP-2000 is used. It is mentioned in the revised manuscript. The proposed article
Naderpour, H.; Mirrashid, M.; Evaluation and verification of finite element analytical models in reinforced concrete members. Iranian Journal of Science and Technology - Transactions of Civil Engineering. 2020, 44, 2, 463-480.
has been included in the reference list.
3- Please define each of the states S1 to S5 that have been considered in stress/deformation graph for the strut model (see Fig. 4).
The states S1 to S5 are defined in section 3.3 of the revised manuscript.
4- Adding two new sections would be helpful: research significance (after introduction), and discussion (before conclusion). In the "Research Significance" section, explain the novelty and research significance in one or two paragraphs. In the "Discussion" section, tell us about the pros and cons, benefits and also limitations of the proposed
The proposed sections are added in the revised manuscript.
5- Please provide figures of full analytical frame within the article. Moreover, please add full details about these two frames.
The full frame is illustrated in Figure 5 of the revised manuscript.
Additional information about the frames are provided in section 3.3 (concerning the materials and modeling issues).
6- It is not clear how the authors determined initial gap between two structures. In this regard, the article “Prediction of critical distance between two MDOF systems subjected to seismic excitation in terms of artificial neural networks” introduce a technique to estimate the gap. Please consider such references in your work.
The results presented in the present paper mainly refer to buildings in contact (dg = 0). Results for separation distances dg > 0 will be presented in a forthcoming paper. Relevant comment is added in “Discussion” section of the revised manuscript. The articles
Naderpour, H.; Khatami, S.M.; Barros, R. Prediction of critical distance between two MDOF systems subjected to seismic excitation in terms of artificial neural networks. Periodica Polytechnica Civil Engineering. 2017, 61, 3, 516-529.
Khatami, S.M.; Naderpour, H.; Mortezaei, A.; Tafreshi, S.T.; Jakubczyk – GaÅ‚czyÅ„ska, A.; Jankowski, R. Predicting the peak structural displacement preventing pounding of buildings during earthquakes. Journal of Physics: Conference Series. 2021, 2070, 1, Article 012010.
have been included in the reference list.
7- Please clarify how the parameters such as hcol, hinf, rinf, were calculated in this study.
A relevant comment is included in section 3.3 of the revised manuscript.
8- The figures presented in section 3, needs to be described and interpreted more.
Additional explanatory comments have been included in section 4 of the revised manuscript (section 3 in the initial version).
9- It is not clear what the black lines are in Fig. 15.
The black lines are the shear force (V) – axial force (N) histories for excitation El Centro – direction +. A relevant clarification is added in Figure 17 of the revised manuscript (Figure 15 in the initial version).
10- Conclusion should be extended.
The manuscript is revised according to the reviewer’s comment.
Reviewer 3 Report
The manuscript examines the pounding cases between structures with different geometry (asymmetric) in both directions (plus and minus) of each seismic excitation. the novel part of the results are interesting that the developing shear forces on the columns that suffer the hit in case of type B pounding exceed the shear strength of the column even if detailing for critical regions to Eurocode 8 are applied. Further, the configuration increases the developing pounding forces and consequently increases the capacity demands mainly in terms of ductility of the column that suffer the hit.Manuscript is written clearly and easy to understand for the readers. I believe that the manuscript can be published as it is.
Author Response
Reviewer 3
The manuscript examines the pounding cases between structures with different geometry (asymmetric) in both directions (plus and minus) of each seismic excitation. the novel part of the results are interesting that the developing shear forces on the columns that suffer the hit in case of type B pounding exceed the shear strength of the column even if detailing for critical regions to Eurocode 8 are applied. Further, the configuration increases the developing pounding forces and consequently increases the capacity demands mainly in terms of ductility of the column that suffer the hit. Manuscript is written clearly and easy to understand for the readers. I believe that the manuscript can be published as it is.
We would like to sincerely thank the reviewer for his encouraging comments and his efforts.
Reviewer 4 Report
The manuscript describes relevant research related to the seismic behavior between two buildings with the potential of pounding between them.
Suggestions:
It is convenient to include more data about the seismic action; for instance, the seismograms of the earthquakes used and their respective pseudo acceleration response spectra. It is useful to explain why you use only 16 secs of the ground motions.
It is helpful to include the fundamental period of each studied building as a basic reference to understand the seismic response.
In order to be able to affirm “that pilotis configuration significantly increases the maximum first floor story drift due to the pounding” it is necessary to compare the story drift of the same building in two cases: a) the seismic response of the building without pounding (this graph is not included in the manuscript), b) the seismic response of the building with pounding (this graph is already included in the manuscript).
Author Response
Reviewer 4
The manuscript describes relevant research related to the seismic behavior between two buildings with the potential of pounding between them.
First of all, we would like to thank the reviewer for his constructive and helpful suggestions which are adopted in the revised manuscript. Here are some clarifications concerning each one of them:
Suggestions:
It is convenient to include more data about the seismic action; for instance, the seismograms of the earthquakes used and their respective pseudo acceleration response spectra. It is useful to explain why you use only 16 secs of the ground motions.
The seismograms and the pseudo-acceleration response spectra of the excitations used in the study are illustrated in Figures 8 and 9 of the revised manuscript.
Full accelerograms are used for the inelastic dynamic analysis. However, the results are presented for the first 16 seconds only just for the sake of clarity of the figures. This time interval always includes the peak response of the structures. A relevant statement is added in “Considered seismic excitations” subsection.
It is helpful to include the fundamental period of each studied building as a basic reference to understand the seismic response.
The fundamental periods of the buildings are reported in “Simulation of interacting multistory structures” subsection of the revised manuscript.
In order to be able to affirm “that pilotis configuration significantly increases the maximum first floor story drift due to the pounding” it is necessary to compare the story drift of the same building in two cases: a) the seismic response of the building without pounding (this graph is not included in the manuscript), b) the seismic response of the building with pounding (this graph is already included in the manuscript).
Figures 18 and 19 in “Discussion” section of the revised manuscript illustrate the comparison between buildings with and without pounding for indicative pounding cases.
Round 2
Reviewer 2 Report
The paper can now be accepted.